# Plasmonic Near-Infrared Photoconductor Based on Hot Hole Collection in the Metal-Semiconductor-Metal Junction

**DOI:** 10.3390/molecules27206922

**Published:** 2022-10-15

**Authors:** Zhiwei Sun, Yongsheng Zhong, Yajin Dong, Qilin Zheng, Xianghong Nan, Zhong Liu, Long Wen, Qin Chen

**Affiliations:** 1Institute of Nanophotonics, Jinan University, Guangzhou 511443, China; 2College of Life Science and Technology, Jinan University, Guangzhou 510632, China

**Keywords:** plasmonic, hot carriers, photoconductor, photodetection, metal-semiconductor-metal

## Abstract

Harvesting energetic carriers from plasmonic resonance has been a hot topic in the field of photodetection in the last decade. By interfacing a plasmonic metal with a semiconductor, the photoelectric conversion mechanism, based on hot carrier emission, is capable of overcoming the band gap limitation imposed by the band-to-band transition of the semiconductor. To date, most of the existing studies focus on plasmonic structural engineering in a single metal-semiconductor (MS) junction system and their responsivities are still quite low in comparison to conventional semiconductor, material-based photodetection platforms. Herein, we propose a new architecture of metal-semiconductor-metal (MSM) junctions on a silicon platform to achieve efficient hot hole collection at infrared wavelengths with a photoconductance gain mechanism. The coplanar interdigitated MSM electrode’s configuration forms a back-to-back Schottky diode and acts simultaneously as the plasmonic absorber/emitter, relying on the hot-spots enriched on the random Au/Si nanoholes structure. The hot hole-mediated photoelectric response was extended far beyond the cut-off wavelength of the silicon. The proposed MSM device with an interdigitated electrode design yields a very high photoconductive gain, leading to a photocurrent responsivity up to several A/W, which is found to be at least 1000 times higher than that of the existing hot carrier based photodetection strategies.

## 1. Introduction

As a key component of every optoelectronic system, the photodetector in the front end of the optical receiver is capable of converting an incoming optical signal into an electrical signal. During the past decades, detection of the near-infrared (NIR) photons has gained increasing importance in modern science and technology due to its widespread applications in diverse areas, including surveillance, telecommunication, motion detection, and biological image sensing [1,2,3]. Most commercial photodetectors are photovoltaic-type devices that are based on crystalline inorganic semiconductors. It is well-established that the photons with energy less than the band gap cannot be absorbed or converted by semiconductors, resulting in a distinct cut-off wavelength. For wavelengths longer than 1100 nm, the fast and efficient detection of NIR light mainly relies on the low-band gap semiconductor system, such as indium gallium arsenide (InGaAs) [4], germanium on silicon [5], mercury cadmium telluride (HgCdTe) [6] and 2D materials [7,8], which requires complex and costly processes, such as molecular beam epitaxy (MBE) and ion implantation. In particular, in pursing low-cost and a high pixel density image sensor, obstacles regarding the incompatibility with existing large-scale foundry technology and signal processing integrated electronic circuits are becoming more and more crucial in these infrared semiconductor-based platforms.

To this end, extensive efforts have been devoted to the development of silicon-compatible or all-silicon NIR photodetection strategies. In order to obtain a sub-band gap photoresponse (λ > 1100 nm), several mechanisms enabling sub-band gap photoresponse, e.g., bulk/surface defect-mediated absorption, two-photon absorption and hyper-doped silicon have been proposed previously for all-silicon-based NIR photodetectors without extra process complexity or cost for introducing foreign NIR materials on silicon [9,10]. However, their practice and implementation are still hampered by a series of problems, including low quantum efficiency, slow response speed, poor reliability, and so on. The internal photoemission effect (IPE), based on the ultrathin metal-silicides (PtSi, NiSi, TiSi_2_, and etc.), was also proposed to obtain NIR photodetection on *p*-type silicon [9]. The IPE photodetectors offer advantages like extremely high switching speed and complete CMOS compatibility, but with the disadvantage of requiring a cryogenic cooling operation. It becomes more difficult to detect NIR wavelengths at room temperature based on IPE [11]. The reported external quantum efficiency (EQE) is low compared to that of detectors that are based on inter-band absorption, and this limits its application, both in power monitoring and in the telecommunication field.

Since the early 2010s, the plasmonic community has developed a very similar NIR photodetection mechanism to IPE that is also based on metallic absorption, but the photo-carriers emitted at the M-S junction are highly energetic, with assistance from the plasmonic resonance and non-radiative decay processes [12,13,14,15,16,17]. These photodetectors called plasmonic hot carrier (hot electron or hot hole) photoconductors and typically have a surface plasmon activated M-S junction system where the metal nanostructure sitting on the semiconductor serves as an absorber and carrier emitter [12]. In the last decade, tremendous efforts have been focused on the optimization of plasmonic absorber designs and construction of the M-S junction with sandwiched or embedded metal for more preferred omni-directional hot carrier emissions. Unfortunately, carrier transport and ejection in metals are relatively inefficient processes compared to semiconductor materials, and previously reported EQE of these hot carrier-based free-space photodetectors are still far below 1% at the telecom wavelengths [12,13,14,15]. In the large family of semiconductor photodetection devices, the photoconductors with a electrical gain effect are often used to achieve infrared light detection [18,19,20,21,22]. The photoconductive gain in these devices can be substantially larger than unity, which means that the obtained photocurrent for a given light power (i.e., the responsivity) can be much larger than that of a junction-based photodetector. In this paper, we propose to utilize the concept of photoconductive gain in the plasmonic hot carrier detector to obtain a much higher photocurrent responsivity at the NIR wavelength regime, which has not yet been investigated in experiments. Relying on the plasmonic hot spot-enriched Au/Si nanoholes (SiNHs) platform, the coplanar MSM interdigital electrodes are used as optical absorbers and simultaneously the electrical collectors for the emitted hot holes at the M-S junction.

## 2. Plasmonic Hot Hole Based MSM Photoconductor

The proposed a plasmonic photoconductor design and its energy band diagram is schematically illustrated in Figure 1. Briefly, the device is fabricated on the SOI (Silicon-On-Insulator) substrate where the thin *p*-type device layer (thickness ~340 nm, 1–20 Ω·cm) has a randomly structured nanohole-based surface morphology. The thin Au interdigital electrodes has a thickness of around 20 nm and is assumed to have a conformal coating on the SiNHs, which forms the broad-band plasmonic absorber with high density hot spots-enabled light absorption. With a thickness smaller than the mean free path of carriers (several tens of nanometer for gold), the transport loss of the photogenerated carrier in the absorber can be efficiently alleviated. The mismatch between the work function of Au and the Fermi level of the *p*-type silicon allows the two interdigitated electrodes to form a back-to-back MSM junction in contact with the thin-film silicon [23]. Upon light illumination, energetic hot electron-hole pairs are generated in the thin Au electrodes and only the hot holes can overcome the M-S barrier and efficiently emit to the valance band of silicon. The hot electrons, however, encountering a large conduction band offset will have a much lower emission probability in our Au/*p*-Si junction. Sun et al. [15] reported a comparison between hot electron and hot hole collection in Au/Si systems (Au/*n*-Si and Au/*p*-Si, respectively) and revealed that the hot hole-based NIR photodetection is more efficient due to the lower emission barrier and more favorable energy distribution upon excitation. Note that, similar to any other semiconductor-based photoconductive device, the proposed plasmonic MSM photoconductor is only functional at the voltage-biased conditions, as shown in Figure 1b,c. Under zero-voltage bias, the flows of the emitted hot holes from two electrodes are equal and will be canceled out. In contrast, with a positive voltage biasing on one of the electrodes, the emitted hot holes will move toward the other electrode and form a constant drift carrier flow, thus leading to a detectable photocurrent output.

## 3. Fabrication of the Au/SiNHs Structure and the MSM Photoconductor

The optical design of the broad-band plasmonic absorber structures is based on our previous study [14]. As shown in Figure 2a, the low-cost, large area and lithography-free fabrication of silicon nanostructures can be achieved through the combination of the simple metal thermal dewetting and metal-assisted chemical etching (MACE) processes. First, ultrathin Au film was deposited on the SOI substrate using the ion beam sputtering method. Second, the Au film covered SOI samples were then subjected to a rapid thermal annealing (RTA) furnace. To form the densely packed gold nanostructures during the thermal dewetting, the optimized RTA conditions are found to be 400 °C and a heating duration of 2 min. The resulting Au nanostructure shown in Figure 2b is in nano-island morphology with sub-wavelength feature sizes. Third, the gold nano-islands were used as the etching seeds in the MACE process. This allows the morphology of the thermal dewetted Au structures to transfer into the underlying silicon substrate. As shown in the middle plot of Figure 2b, the Au islands were embedded in the silicon. We removed these Au nano-seeds using gold etchant and subsequently the residual oxide in buffered hydrofluoric acid. Finally, a 20 nm thick Au film was deposited on the structured silicon via magnetron sputtering. The fabricated Au/SiNHs plasmonic structure is presented in the bottom of Figure 2b, which shows a complex random morphology that is essentially complimentary to the initial thermal dewetted Au nano-islands. According to our pervious study, such densely packed Au/SiNHs have optimum lateral morphologies that sustain hot spots and extend over the mesoscopic dimensions; therefore, it is more favorable for plasmonic light trapping in a broad wavelength regime [14].

Figure 2c shows the device fabrication flow for our proposed MSM photoconductor. In consideration of the electrical isolation, a 50 nm thick alumina (Al_2_O_3_) by atomic layer deposition (ALD) was prepared in the structured SOI sample. The active device area was defined by the selectively etched window opening on the alumina layer. Then, the interdigitated electrodes were fabricated by standard UV lithography, metal deposition (e-beam evaporation, 20 nm thick Au film) and lift-off processes. Additionally, thick wiring electrodes were also fabricated in contact with the thin interdigitated MSM contacts for the sake of obtaining more reliable electrical connection and decreased series resistance. The complete plasmonic MSM photoconductor samples were mounted on a PCB board to ease the electrical and optoelectrical measurements. Wire bonding is then adopted to connect the thick electrode pads of the plasmonic device with the electric leads of the PCB.

## 4. Plasmonic Absorption Properties of the Au/SiNHs Structure

The fabricated Au/SiNHs are closely and randomly packed nanostructures, which involve a fairly large amount of narrow nano-gaps with the sub-wavelength size enabling strong electric field localization. Such a hot spot-enriched random plasmonic structure offers high light absorption in a broad-band wavelength regime. In order to quantitatively evaluate its optical properties, as shown in Figure 3a, three-dimensional Finite-Difference Time-Domain (FDTD)-based numerical simulations were carried out by incorporating realistic morphologies of the Au/SiNHs into the model. The calculated light absorption spectra of the plasmonic Au/SiNHs and a planar Au/Si reference structure were presented in Figure 3b. Note that, at the wavelength region of 1100 to 2000 nm, the absorption loss in silicon can be neglected. The planar reference structure (20 nm Au film on SOI without nanostructures) has very low light absorption (below 10%) due to the substantial reflection losses. There are two small fluctuations that appeared in the curve and they can be safely attributed to the optical interferences of the SOI layer structures. In contrast, the plasmonic Au/SiNHs exhibits high light absorption ability (approximate to 82% at the peak) over a broad wavelength regime.

The electric field distributions at three selected wavelengths are presented in Figure 3c, to explore the physical origins of the observed broad-band plasmonic feature of random Au/SiNH structures. For all the three wavelengths, due to the random nature of the plasmonic Au/SiNHs, there are a large number of hot spots (high electric field localization) in those areas with a narrow void or shape tips, which collectively contribute to the broad-band plasmonic absorption, as illustrated in Figure 3b. In particular, for the wavelength of 1300 nm, the overall electric field intensity is much higher than that of the other two wavelengths, thus giving rise to higher light absorption at this wavelength. In Figure 3c, we marked the blue frame in the electric field distribution at 1300 nm, which represents a small local area composed of hot spots with the highest electric field concentration. The corresponding normalized light absorption (i.e., the total power absorption obtained by integrating the losses over the whole volume) is plotted as a blue line in Figure 3b. Interestingly, the absorption spectra of this small area exhibit a distinct and narrow peak at 1300 nm, revealing a more well-defined plasmonic resonance characteristic. Therefore, we believe that the efficient and broad-band light absorption observed in our random Au/SiNHs structure mainly stems from the field enhancement effects of the localized surface plasmon resonances generated at different areas. Based on such a hot-spots enriched plasmonic system, the high density of energetic hot carriers can be generated in the ultra-thin Au layer and subsequently collected by the MSM electrodes.

## 5. Photoelectric Characterization on the Plasmonic MSM Photoconductor

Figure 4a shows the plasmonic hot hole photoconductor with a pair of interdigitated MSM electrodes. Four different MSM electrode designs (D1 to D4) were selected for our photoelectric characterization. They have different sizes of the width of the electrode fingers (defined as W) and fixed separation (S = 4 μm) between two fingers. The corresponding dark I–V curves are also plotted in Figure 4a, which are almost anti-symmetrical for all four devices. With the increasing electrode width, the dark currents at positive or negative voltages increase due to the increment of the effective area of the M-S contact. Note that the current amplitudes at positive and negative biases are not exactly the same because of the difference between the two wiring electrodes. The photoelectric response of the photoconductor were characterized based on the high-power supercontinuum white laser system, which is equipped with a NIR AOTF (both from NKT Photonics) allowing the output monochromatic light with sufficient high power (10~20 mW, spot size of 2~3 mm in diameter). All the electrical characteristics were collected using a SourceMeter (Keithley 2636B).

The time dependence of the photocurrent with repeatedly switching on/off of the laser illumination (the wavelength is swept continuously from 1600 to 1100 nm with a step of 25 nm) is depicted in Figure 4b. To obtain the photoelectric response, the MSM photoconductors were measured under external voltage biasing conditions (*V*_b_ = 4, 6 and 8 V). As compared to the plasmonic device, the planar reference MSM device with a pure silicon response has significant lower photocurrents throughout the investigated wavelength regime and appears to be completely cut off for the wavelengths larger than 1200 nm. Obviously, the plasmonic based device is capable of efficiently detecting NIR light in a broader range of wavelengths. The output current rises dramatically and immediately when the laser is turned on, and decreases fast to the dark current level once the illumination is switched off. As indicated by the insert plot, the photoresponse induced offsets of the plasmonic device can be clearly distinguished even at a wavelength of 1600 nm. Note that the large photocurrents observed near 1100 nm could be safely attributed to the intrinsic silicon response, which shows a cut-off wavelength of around 1200 nm in our case. Interestingly, there exists a distinct response peak at a wavelength of 1300 nm, pointing to a plasmonic resonance induced photoelectric conversion mechanism. As we have shown previously in Figure 1, for the M-S junction on the *p*-type silicon, only plasmonic hot holes can overcome the Schottky barrier and thermally emit to the valance band of silicon. Hence, we believe that the observed photoresponse beyond 1200 nm is due to the photoelectric conversion of the plasmonic hot holes.

As shown in Figure 4b, the plasmonic MSM photoconductor exhibits a voltage-dependent response characteristic as the photocurrents increase monotonically with an increasing voltage bias, indicating that there exists a photoconductive gain mechanism. The proposed device scheme is based on the combination of plasmonic hot carrier collection and the photoconductive gain mechanisms. The conductivity enhancement of the silicon channel between the two electrodes upon sub-band gap light illumination mainly arises from the ejection of the plasmonic hot holes through the M-S junctions. This is quite different from the case of a conventional semiconductor-based photoconductor where the excess carriers are generated directly in silicon at the areas between the two electrodes, if the photon energy is larger than the band gap of silicon. With the increment of electrical conductivity via hot hole emission, the proposed device with voltage-bias dependent photocurrent response shown in Figure 4b therefore points to a clear photoconductive gain effect. In Figure 4c, in order to reveal the performance of the operation speed, we inserted an optical chopper into the light path and detected the temporal current response of the plasmonic MSM photoconductor. It is well established that the hot carrier dynamics (like generation, transport and emission) for photoelectric conversion are ultra-fast processes, rendering the device capable of high-speed operation. This is confirmed by the chopping frequency-dependent response shown in Figure 4c. For the largest chopping frequency of 1 kHZ, the photocurrent response of the MSM photoconductor remains high, with an amplitude approaching 70% of the slow modulated results (e.g., 100 HZ). It is apparent that the demonstrated Au-silicon MSM photoconductive device is capable of tracking the fast-varying optical signal with a millisecond response time.

The spectral photocurrent responsivities of the MSM photodetectors (D4) under different biasing voltages are presented in Figure 5a. It is observed that the planar reference device exhibits poor performance with measured responsivities significantly lower than that of the nanostructured one. Relying on the photovoltaic mechanism, the response of planar device becomes negligible for the wavelengths with photon energy smaller than the band gap of silicon. Apart from the silicon response near 1100 nm, an additional resonance peak associated with the plasmonic mechanism can be found in the spectral responsivity of the MSM photoconductor. The observed peak is located at a wavelength of about 1300 nm, which is in good agreement with the simulation results depicted in Figure 3b,c. With an increasing voltage bias, the photocurrent responsivity increases gradually and appears to be saturated at 8 V. The dependence of photocurrent response to external biasing arises from the increment of the photoconductive gain. The spectral responses of plasmonic MSM photoconductors with different electrode width are presented in Figure 5b. For the device with the smallest electrode width, the peak responsivity is found to be 5.12 A/W. This value is at least 1000 times larger than that of the existing hot carrier devices that operates at the same wavelength regime [12,13,14,15,16,17], thus suggesting the existence of a photoconductive gain effect in our proposed MSM device.

## 6. Conclusions

We have proposed to incorporate the photoconductive design into the plasmonic hot carrier-based device to obtain more efficient near-infrared photodetection on the silicon platform. The device with a coplanar metal-semiconductor-metal (MSM) interdigital electrodes differs from the classic photoconductor mainly in that its light absorption takes place in the metal electrodes rather than in the semiconductor. Relying on the hot-spots enriched random Au/Si nanohole structure, the proposed plasmonic photoconductor exhibits broad-band light absorption behavior. The photogenerated hot holes with energy larger than the Schottky barrier can be emitted over the barrier, which forms a directional photocurrent flow between the biased MSM electrodes. Due to the existence of a large photoconductive gain effect, for the device with optimized electrode geometry, the photocurrent responsivity at the resonance peak is found to be 5.12 A/W, which is at least 1000 times larger than that of the existing hot carrier devices that operate at same wavelength regime.

## Figures and Tables

**Figure 1 molecules-27-06922-f001:**
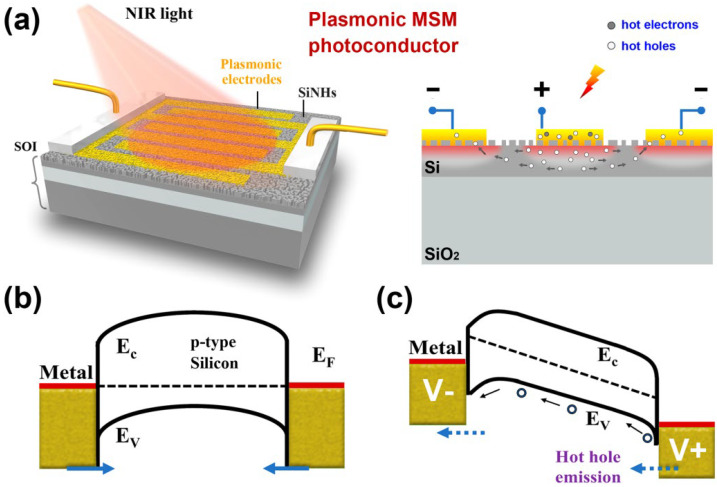
Plasmonic MSM photoconductor based on random Au/SiNHs. (**a**) Schematic drawing and work principle of the device. (**b**,**c**) Energy band diagrams of the MSM photoconductor under zero-bias and bias conditions.

**Figure 2 molecules-27-06922-f002:**
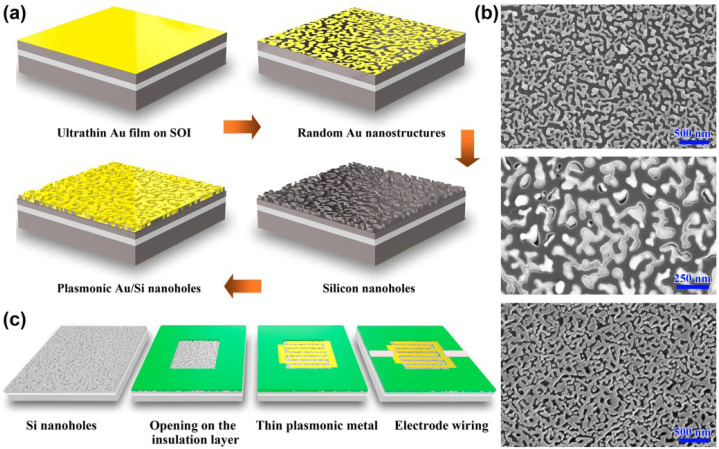
Structure and device fabrication processes of the plasmonic MSM photoconductor. (**a**) Lithography-free fabrication of the Au/SiNHs nanostructures. (**b**) Au nanostructures by the thermal dewetting method (**top**), MACE etched Si with (**middle**) and without (**bottom**) Au etching seeds. (**c**) MSM device fabrication flow.

**Figure 3 molecules-27-06922-f003:**
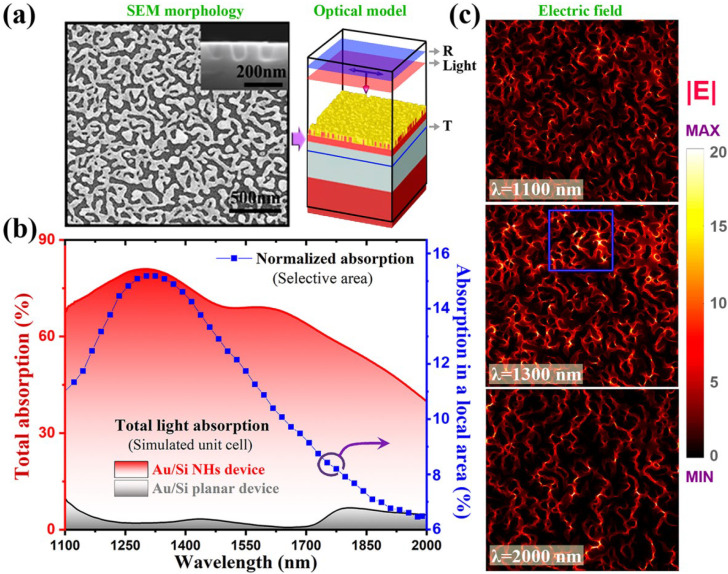
Three-dimensional optical modeling of the light absorption property of the Au/SiNHs structures. (**a**) Combining realistic morphologies with FDTD simulation. The simulated area is 2.5 × 2.5 μm^2^. (**b**) Simulated absorption spectra of the nanostructured and planar Au/Si device. (**c**) Normalized electric filed distributions at three wavelengths. A hot spot area with high field intensities is marked with a blue frame and its corresponding absorption is plotted in (**b**).

**Figure 4 molecules-27-06922-f004:**
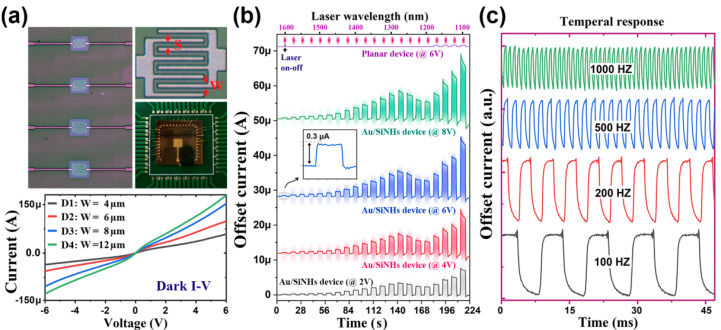
(**a**) Microscope images and typical dark I–V curves of the MSM devices. (**b**) Wavelength and time-dependent photoresponse of the Au-SiNH device (D4) under different voltage biasing conditions. The inset is the enlarged plot of the low photoelectric response regime. To ease the comparison, the result of the planar reference device biased at 6 V is also given in the plots. (**c**) The temporal current response for the device (D4) illuminated by the modulated incidence with different chopping frequencies.

**Figure 5 molecules-27-06922-f005:**
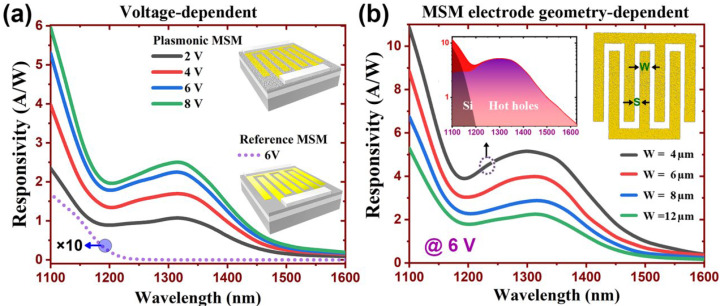
The measured spectral photocurrent responsivity of the plasmonic and planar MSM photoconductors. (**a**) Voltage-dependent responsivities for the device with a fixed electrode finger width (D4, W = 12 μm). (**b**) The measured responsivities for devices with four different electrode geometries.

## Data Availability

Not applicable.

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
