# Peer review of "Plasmonic Near-Infrared Photoconductor Based on Hot Hole Collection in the Metal-Semiconductor-Metal Junction"

_molecules, 2022, doi:10.3390/molecules27206922_

Round 1

Reviewer 1 Report

Dear Editor

Thank you for sending the manuscript. Overall, the manuscript seems important and interesting, well presented. My opinion is the manuscript may be considered for publication in Molecules.

Author Response

We sincerely thank you for your feedback and postive comments.

Reviewer 2 Report

This paper proposes a new structure of metal-semiconductor-metal (MSM) junction on silicon platform to realize efficient hot hole collection and photoconductive gain mechanism of infrared wavelength. This structure is at least 1000 times larger than the existing heat carrier devices operating at the same wavelength system. This article is a valuable work, but it needs to be modified properly. Here are some suggestions:

1. The legend position of the whole paper needs to be adjusted;

2. The analysis of the working principle of MSM photoconductor is not detailed enoughï¼›

3. Why use Au/SiNHs structure? What is the better research significance of this structure compared with other material structures?

4. It is suggested that a thermal reference value be given in Figure 3 (c), that coordinate values be given for the right vertical axis in Figure 3 (b), and that there is an error in the format of the coordinate area units in the Figure 3 figure notes.

5. I think the following paper may be of great reference value for improving your manuscript. (Small 2019,15(38),1902811ï¼›Particle & Particle Systems Characterization, 1800341, 2018ï¼›ACS Applied Materials & Interfaces, 12(38), 43098-43105, 2020ï¼›Applied Materials & Today, 2022,28,101546)

Reviewer 3 Report

Authors reported a plasmonic near-infrared photoconductor using a MSM structure.

Some comments and a concern are:

1. Could you please clarify the motivation why the authors focus on hot hole, not hot electron. material selection, mean free path, EQE etc.

2. L69. "carrier transport and ejection in metals are relatively un-efficient processes compared to semiconductor materials"

Could you please explain the reason, and your approach to solve it? As introduction mention the low external quantum efficiency (EQE) in conventional devices, I am wondering whether the authors study the EQE in this paper.

3. How the random Au nanostructures affect carrier transport? Surface roughness scattering of electrons are one of the issues in bulk MOSFET. Will there be disadvantage in terms of carrier transportation in this structure.   

4. L111. "To form the densely packed gold nanostructures during the thermal dewetting, the optimized RTA conditions are found to be 400 ℃ and heating" 

Could you please comment on why densely packed is needed if not yet, and whether you have the images to show how different if change the condition.

5. In the Fig.2, Fig.5 how the authors chose the separation distance and thickness of Au film? 

6. Authors used the e-beam deposition to form the electrode wiring. Did you have your sidewall of the silicon nanoholes also being covered with Au or not?

7. L191. "equipped a NIR AOTF (both from NKT Photonics) allowing output monochromatic light with sufficient high power (several mW)." Do authors have some numbers. 

Overall, I think the manuscript is well organized, however, I think some points need to be clarified.
